# The Association between Nursing Skill Mix and Patient Outcomes in a Mental Health Setting: Protocol for an Observational Feasibility Study

**DOI:** 10.3390/ijerph19074357

**Published:** 2022-04-05

**Authors:** Nompilo Moyo, Martin Jones, Shaun Dennis, Karan Sharma, Richard Gray

**Affiliations:** 1School of Nursing and Midwifery, La Trobe University, Melbourne, VIC 3086, Australia; martin.jones@unisa.edu.au (M.J.); r.gray@latrobe.edu.au (R.G.); 2Victorian Tuberculosis Program, Melbourne Health, Melbourne, VIC 3000, Australia; 3Department of Rural Health, University of South Australia, Whyalla Norrie, SA 5608, Australia; 4IIMPACT in Health, University of South Australia, Adelaide, SA 5000, Australia; 5Whyalla Integrated Mental Health Service, Flinders & Upper North Local Health Network, Whyalla, SA 5600, Australia; shaun.dennis@sa.gov.au; 6Austin Health, Melbourne, VIC 3084, Australia; karan.sharma@austin.org.au

**Keywords:** readmission, mental health nurse, inpatient, nurse skill mix

## Abstract

International research on nursing skill mix has focused primarily on medical and surgical patient populations. The association between nursing skill mix and clinical outcomes for psychiatric inpatients has not been explored. The aim of this study is to establish the feasibility of extracting and linking nurse and inpatient data from hospital administrative data sources. This is an observational study. Data will be extracted from hospital administrative sources and linked together. Patient information will include duration and number of psychiatric hospital admissions. We will extract information on the educational preparation of nurses working in the participating hospitals to enable us to calculate estimates of the nursing skill mix. The study will be conducted in two psychiatric inpatient services in Australia. Our study will test the feasibility of extracting and linking nursing skill mix and patient data in a mental health setting and will inform the methodological development of an appropriately powered observational study. **Australian and New Zealand clinical trial registry:** ACTRN12619001337167p.

## 1. Introduction

### 1.1. Mental Health Nursing Workforce

There is considerable variation in the skill mix of nurses working in mental health settings. Some countries have a blend of registered nurses and those with specific mental health nurse education, this is the model in Australia [1]. In other countries, such as the United Kingdom, wards are staffed entirely by mental health nurses [2].

The World Health Organisation (WHO) estimates that in 2020 there were 3.8 nurses per 100,000 people working in mental health settings globally [3]. The WHO regions and income groups differed widely. For example, the Western Pacific Region had 5.3 nurses per 100,000 people, while the European Region had 25.2 nurses per 100,000 persons. Similarly, low-income nations had 0.4 nurses per 100,000 people, whereas high-income countries had 29 nurses per 100,000 [3]. It’s worth noting that the WHO makes no mention of the number of nurses who are specifically trained in mental health nursing. In 2019, Australia had 90.2 full-time equivalents (FTEs) mental health nurses per 100,000 population [4].

### 1.2. Nursing Education and Patient Outcomes

The association between nursing education and patient outcomes (mortality, medication errors) in medical and surgical settings has been extensively studied [5,6,7,8]. Most authors report that a more highly qualified nursing workforce is associated with lower odds of mortality. For example, Lasater et al. [8] conducted an observational study using data from two cross-sections in time (2006, 2016) to examine if changes in hospital employment of nurses with bachelor’s degrees over time were associated with differences in patient outcomes in surgical wards of 519 hospitals. The authors used a nurse survey, American Hospital Association Annual Survey, and patient administrative claims to collect data. Patients in hospitals with a higher proportion of nurses with bachelor’s degrees had lower risk-adjusted mortality, 7-day and 30-day readmission rates, and shorter hospital stays [8].

### 1.3. Mental Health Nursing Education

The definition of a mental health nurse differs from country to country. For example, in the United States of America and Australia, nurses complete a “comprehensive” qualification that would enable them to work in any clinical setting, including mental health [9,10,11]. Nurses may then go on to complete specialist mental health nurse education [10,11].

In the United Kingdom, there is a direct pathway into mental health nursing. Registrants in this field of practice are only licenced to work clinically in mental health settings [12]. Because of the nursing skill mix–comprehensive nurses and nurses with postgraduate education-in the Australian mental health system, there is the potential to calculate a comprehensive to mental health nurse ratio.

### 1.4. Mental Health Nurse Education and Patient Outcomes

We have previously reported a systematic review of observational and experimental studies that tested the association between mental health nurse skill mix and patient outcomes (relapse of illness) in a psychiatric inpatient setting [13]. We found no studies that tested this association, suggesting a gap in knowledge. Prior to undertaking a full-scale appropriately powered observational study, there is a need to establish the feasibility of undertaking such research.

### 1.5. Outcomes in Mental Health Skill Mix Research

Mortality is typically used as an outcome in skill mix research in medical and surgical settings. The use of mortality as an outcome in mental health is problematic. Death in inpatient settings is a rare event [14]. Consequently, it should not be considered a meaningful or practical outcome for use in mental health skill mix research.

The need for further inpatient admissions is common for people experiencing mental-ill health. For example, in one multi country study, more than 15% of people with schizophrenia required readmission to psychiatric inpatient units within 30 days of discharge [15]. Relapse of mental-ill health is the most common–but not only–reason why people are admitted for psychiatric inpatient care [16]. In theory, admission to hospital is a valid and meaningful outcome measure that is routinely recorded by mental health services.

### 1.6. Aim

The aim of this study was to test the feasibility of extracting and linking administrative psychiatric inpatient and nurse data to test the association between the mental health nurse to comprehensive nurse ratio and patient readmission within 12 months of discharge. If feasible, we will develop a protocol for an appropriately powered observational study.

## 2. Materials and Methods

This is an observational data linkage study. We have structured our manuscript according to the Strengthening the Reporting of Observational Studies in Epidemiology (STROBE) guidelines (see Appendix A). Our study was registered with the Australian and New Zealand clinical trial registry: ACTRN12619001337167p on 30 September 2019.

Most previous studies have determined skill mix (e.g., the nurse-to-patient ratio) using a self-report questionnaire, e.g., [8] that asks nurses to estimate the number of nurses and patients present during their last shift. The validity of this measure of skill mix has not been established and is likely subject to multiple sources of bias (social desirability, non-response). The approach we are testing in this study relies on linking data from administrative datasets together, which may be a more valid approach to estimating skill mix.

### 2.1. Study Setting

Our feasibility study will be conducted in two participating settings, one in South Australia (rural) and the other in Victoria (metropolitan). The site in South Australia is the only rural health care provider in South Australia with a specialised inpatient mental health facility and mental health rehabilitation service. The psychiatric inpatient unit has six beds.

The site in Victoria is a major public health provider in Melbourne. The site has two adult acute inpatient wards and provides a range of community mental health services (for example, mobile outreach service and case management). We chose two sites to test if our study is feasible in different states of Australia as well as in rural and metropolitan areas.

### 2.2. Participants

We will retrospectively extract administrative data on adult psychiatric inpatients diagnosed with any mental disorder who were inpatient for at least 24 h. Data on nurses working in the participating clinical service during the study period will also be extracted. Patients and nurses will be included in the study if they meet the following inclusion criteria.

### 2.3. Inclusion Criteria (Patients)

Aged 18 years or older.Diagnosed with any mental disorder.Admitted as a psychiatric inpatient for at least 24 h from 6 January 2020 to 5 March 2020.

### 2.4. Inclusion Criteria (Nurses)

All registered first level, second level nurses, nursing assistants and student nurses working on participating mental health inpatient units across all shifts (morning, afternoon, and night) from 6 January 2020 to 5 March 2020.

### 2.5. Data Sources

Participating hospitals have different patient and human resource administration systems. Consequently, we anticipate that slightly different procedures will be required to extract and code the data:○Patient-level data:
■South Australian hospital (Integrated South Australian Activity Collection (ISAAC)).■Victorian hospital (Cerner).○Nurse-level data:■South Australian hospital (Complete Human Resource Information System [Chris21] and paper shift roster).■Victorian hospital (Kronos and paper roster).

### 2.6. ISAAC

ISAAC is an electronic patient administration system that records details of public psychiatric inpatient hospitalisations [17]. Data are entered by hospital staff throughout the period of hospitalisation.

### 2.7. Cerner

The electronic patient record system used by the Victorian hospital is called Cerner and is an extensively used system.

### 2.8. Chris21

Chris21 is a self-service payroll and human resource software package that allows managers, human resources personnel and employees to enter timesheets, review and request leave [17,18].

### 2.9. Kronos

Kronos is a suite of workforce management tools [19]. Employee demographic information, hours worked, and periods of leave are all recorded in the system [19].

### 2.10. Patient-Level Data

From the relevant electronic patient records, we will extract and code (details of how we will code in parentheses) the following variables: Age (age in years at point of hospital admission), gender (coded: 1. Male, 2. Female, 3. Other), employment status at the point of admission (coded: 1. Employed, 2. Unemployed, 3. Student, 4. Retired, 5. Other), primary psychiatric diagnosis (coded: 1. Mood and/or anxiety disorder [including bipolar], 2. Schizophrenia, 3. Personality disorders, 4. Other [specify]), psychiatric comorbidities (coded: 1. Yes, 2. No), substance use (coded: 1. Yes, 2. No), physical comorbidities (coded: 1. Yes, 2. No), date of admission (dd/mm/20yy), detained under mental health law (coded: 1. Yes, 2. No), on a Community Treatment Order prior to admission to hospital (coded: 1. Yes, 2. No), when was the patient discharged from the participating ward? (dd/mm/20yy), Discharged on a Community Treatment Order (coded: 1. Yes, 2. No), admitted by inpatient psychiatric services at the participating hospitals within 12 months of discharge (coded: 1. Yes, 2. No), number of psychiatric admissions during the study period, Health of the National Outcome Scales (HoNOS) total score on admission to hospital, HoNOS total score immediately prior to discharge (recorded as number). Health of the Nation Outcome Scales (HoNOS) is a clinician rated tool comprising 12 simple scales assessing behaviour, impairment, symptoms and social functioning for those in the 18–64 years old age group [20]. We will request the unit managers to extract information on the total number of inpatients on the ward at morning handover (07:00 am) each day between the 6 January 2020 and 5 March 2020 (60 days). The patient data extraction form is available as Appendix A.

### 2.11. The Study Period for Patients

The study period for patients is from 6 January 2020 to 5 March 2021. The time frame includes 60 days of inpatient admission (6 January 2020 to 5 March 2021) and 12 months of follow-up period after hospital discharge).

### 2.12. Nurse-Level Data

#### Roster Data

Roster data will be extracted from CHRIS21 and Kronos for the time period 6 January 2020 through 5 March 2020. System Performance Support [21] defined a roster as a timetable that specifies the number of nurses and their classifications required for each shift. Data on the training nurses have received will be extracted and coded: 1. Mental health nurse, 2. Comprehensive nurse, 3. Enrolled nurse, 4. Nursing assistant, 5. Student nurse. Nurse unit managers will be asked to extract and code data from the shift roster about whether nurses have mental health nursing qualifications. We estimate that data extraction and coding should take no more than four hours of a nurse unit manager’s (NUMs) time. We will check with NUMs during the conduct of the study if the tasks were overly burdensome. We defined a mental health nurse as a registered nurse with a formal educational qualification in mental health nursing. A comprehensive nurse is a registered nurse with a formal educational qualification in general/comprehensive nursing and does not hold a formal mental health nursing qualification. An enrolled nurse is a nurse with a certificate or diploma qualification in nursing and is registered as an enrolled nurse. First level nurse refers to mental health and comprehensive nurses. Second level nurses are enrolled nurses. These data will be entered onto a data extraction form available as Appendix A.

We will use the total number of nurses working each day as the denominator and then calculate the ratio of 1. Mental health nurse to comprehensive nurse, 2. Registered nurse to enrolled nurse, 3. Qualified nurse to unqualified staff.

### 2.13. De-Identifying Data

Identifiable data will not be extracted, for example, the name of the hospital, Unit Record Number (URN) of patients, employee number of nurses, names, addresses, date of birth, phone numbers for both patients and nurses.

### 2.14. Data Storage

Data from participating hospitals will be transferred to the participating University in Victoria through the Cloudstor file transfer. CloudStor is a secure online service that was developed by Australia’s Academic and Research Network (AARNet) that enables universities and the wider community to quickly and securely synchronise, share and store files using the high-speed AARNet network. Data will be stored on a password protected CloudStor and only the researchers will have access. Patient and nurse data will be stored in separate files and then combined to enable analysis.

Following completion of the study, all data collected will be stored by the participating University for seven years following the Public Records Act 1973. We do not intend to make these data available to other researchers because this is a feasibility study, and we do not intend to undertake any formal tests of association.

### 2.15. Disposing of Data

Records stored on Cloudstor will be deleted in line with the current Cloudstor process at the time of destruction.

### 2.16. Ethics Considerations

Permission to conduct the research project was granted by the participating University and two Hospitals’ Human Research Ethics Committees: HREC/70480/Austin-2021 and 2021/SSA00366.

We applied for and were granted a waiver of consent because the study was considered by the ethics committee not likely to cause any discomfort or distress to the participants as we will not directly collect data from them and no identifiable information (e.g., names, hospital numbers, addresses, name of the hospital) will be extracted. The anonymity of the participants can, consequently, be always assured.

### 2.17. Data Management

The following procedure for data management will be followed: Data re-coding and checking, removal of duplicates and unwanted data, management of outliers, checking for consistency, and management of missing data. Stata version 14.0 (Stata Corp., College Station, TX, USA) will be used to execute data management procedures.

### 2.18. Data Recoding

Data will be recoded using the following procedure: A codebook will be created that will contain variable names and numeric codes. A number will be assigned to the label for each variable, for example, for gender, male will be coded 1 and female 2, other 3.

### 2.19. Data Checking

Error in data entry may occur when nurses or psychiatrists make clinical entries into the medical record (for example, at the point of admission to hospital). Common errors in data entry often relate to transposition error, such as: Flipping the last two digits of the year of admission (for example, 16 January 2021 might be entered as 16 January 2120). In this study, we will check for the following potential errors in data entry:○Duplicate and unwanted observations○Outliers○Data consistency○Missing data

### 2.20. Removal of Duplicate and Unwanted Observations

We will check if participants have been entered multiple times in the data. For example, the same admission is entered more than once in the data in error. Where we identify a duplicate entry, we will confirm the entries are duplicates by comparing the patient study identification number and date of admission. If they are the same, we will conclude that an error has occurred. We will retain the entry with most data noting that this may mean that we remove valid observations. However, if the researcher “chooses” which data to retain, this would introduce substantial potential for bias because the researcher might be inclined to retain observation favourable to the primary hypothesis [22].

### 2.21. Outliers

Outliers are extreme values that are well outside the overall pattern of a distribution of variables leading to under- or overestimation of the true mean. A potentially important source of bias, it is important to consider in advance how these will be managed in the analysis of the data set [23]. To identify outliers, we will use box plots. Box plots are considered most suitable for exploring both symmetric and skewed data. Infrequent values in categorical data can also be identified using the box plot [23].

There are two main procedures for managing outliers: First, we will go back to the data and check the original value entered. If the extracted number is incorrect, we will amend our database accordingly, documenting any changes that have been made. Second, if the source and extracted data match but the value is seemingly impossible (e.g., the participant was 150 years of age), we will delete the data and consider the value as missing in the analysis.

### 2.22. Data Consistency

Categorical variables and values will be checked for typing errors and inconsistent capitalization, for example, in gender, male can be entered as male in some participants and in some as male or m-ale. Inconsistent values can cause unnecessary delays during the analysis [22,23,24,25]. We will also check that dates are recorded in a consistent way (i.e., day, month, year and not month, day, year).

### 2.23. Management of Missing Data

We will determine if there is a pattern to missing values, are data missing at random or is there a systematic reason for the missingness of data, i.e., missingness depends at least in part on unobserved variables [26].

We will use Little’s MCAR test to assess if data are missing completely at random (MCAR). The data will be assumed to be MCAR if the *p*-value for Little’s MCAR test is not significant. If data are not MCAR, we will conduct further exploratory tests to determine missingness at random (for example, the significance tests of missingness). The significance tests of missingness will determine if data are missing at random (MAR) or missing not at random (MNAR) [26].

The key variables in this study are readmission and the educational preparation of nurses. A listwise deletion method will be used where data are missing completely at random, and the missing data include either of the two key variables. The use of listwise deletion, in this case, is appropriate because data for patients with no outcomes and data for nurses missing educational preparation cannot be analysed to give the association between patient outcomes and educational preparation of nurses.

We will use the pairwise deletion method where data will be missing completely at random and do not involve key variables (i.e., readmission). This will enable us to retain data and minimise the potential for selection bias.

If data are MAR, multiple imputations will be used using three steps: Imputation, completed-data analysis (estimation), and pooling step [27]. We will use listwise deletion if data are MNAR since there is no acceptable remedy for it [26].

### 2.24. Data Analysis

#### Describing the Data

Our final data set will be at an individual patient level. Continuous data (e.g., age, length of stay) will initially be described using means and standard deviations. Categorical variables (e.g., gender) will be reported as numbers and proportions. We will use tables to summarise the characteristics of study participants (nurses and patients).

### 2.25. Calculating Nursing Skill Mix

Each patient will have an individual row in the dataset. For each day (07:00 to 06:59) that the patient is in hospital, we will calculate the total number of registered nurses, unqualified nurses, mental health nurses, comprehensive nurses, enrolled nurses, student nurses, nursing assistants, and total number of patients on the ward. We will use these data to compute the following skill mix ratios on a day-by-day basis:○Mental health nurse to comprehensive nurse.○Qualified nurse to nursing assistant.○All nurse (including nursing assistant) to patient.○Mental health nurse to patient.○Comprehensive nurse to patient.

To compute the final skill mix measure, we will calculate averages for the duration of their admission, using days of admission as the denominator. For example, to calculate the comprehensive nurse to mental health nurse ratio. we would perform the following calculation. A patient was in hospital for three days. On the first day, there were eight comprehensive and two mental health nurses who provided care. On the second and third days, respectively, there were nine and ten comprehensive and three and four mental health nurses. This would give us a skill mix ratio of comprehensive to mental health nurses of 3:1.

To calculate nursing hours per patient day, we will total the number of nurses working on a given day, multiply by seven days and then divide by the number of patients on the ward at midnight.

### 2.26. Inferential Analysis

We will not make inferences from our data because this is a feasibility study.

## 3. Results

Nurse and patient data will be summarised in tables.

## 4. Discussion

We are publishing the protocol for our research to inform other researchers of the work we are doing and the methodological approaches we have adopted, we hope that this will contribute to preventing duplication of effort.

Within an international context, Australia is unusual in operating a comprehensive model of nurse training. The methodology we have proposed can only be conducted in countries–such as Australia–where inpatient mental health services are staffed by a mix of comprehensive nurses and specially trained mental health nurses. We would not be able to replicate our study in the UK, for example, where all nurses must be on the professional register as mental health nurses. It is debatable whether the findings of our research can be generalised beyond an Australian context.

### Limitations

We have not tested the validity and reliability of routine hospital data collection, which is common in this type of research. Routinely collected health data in research studies are accepted as valid and reliable in Australia and worldwide. For example, in the United Kingdom, routine primary care data have been used in research for almost 30 years [28]. Still, the accuracy of the data has never been checked. Hospitals do not collect data on agency nurses’ educational preparation, which may affect the precision of our skill mix estimates. Nursing hours per patient day estimate the total amount of care time available. Likely, it is an overestimate of the actual amount of bedside nursing care patients receive.

We proposed extracting nurse data on gender, age, years of experience, and the type of contract (full-time or part-time), however, the human research ethics committee advised that these data were not required to achieve the aims of the study. It may be important to collect these data in future studies where it will be necessary to adjust for confounding in any proposed analysis. COVID-19-related data, such as isolation, vaccination status, or infections, which could be confounding factors, will not be retrieved. We anticipate that data on COVID-19 would be difficult to obtain.

## 5. Conclusions

This study may provide evidence of the feasibility of extracting routinely collected administrative patient and nurse data to calculate measures of nursing skill mix. Findings from this feasibility study will inform the design and methods for an appropriately powered study of nursing skill mix and patient outcomes in mental health settings. Our study will be the first study to explore the feasibility of using administrative data to calculate the mental health nursing skill mix and patient outcomes.

## Data Availability

The datasets generated during this study will not be publicly available.

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
