# Peer review of "The Association between Nursing Skill Mix and Patient Outcomes in a Mental Health Setting: Protocol for an Observational Feasibility Study"

_ijerph, 2022, doi:10.3390/ijerph19074357_

Round 1

Reviewer 1 Report

The manuscript is well organized and didactic, adequate for its purpose. As the authors say, "the aim of this study is to establish the feasibility of extracting nurse and psychiatric inpatient data from hospital administrative data sources."

The writing style in topics facilitates the identification of the contents covered, although it diminishes the articulation between the literature highlighted to support the proposal. The method was thoroughly described, since it is the most relevant part of this work, given its objective. The study will be developed in the context of Australia, involving nursing professionals (with different levels of professional qualification) and patients from two hospitals, involving urban and rural areas.

The description of the study proposal and its operationalization method seems feasible and may bring relevant information to understand the possible association between the professional qualification in mental health (of the nursing team) and the outcomes of psychiatric hospitalizations, as hypothesized by the authors.

Supplementary materials are adequate and presented in a didactic way, facilitating the understanding of the protocol to be applied in Australia.
In terms of suggestions to the authors, it might be relevant to present their thoughts on the eventual applicability of this protocol to other contexts outside Australia.
Given the above, my opinion is favourable to the publication of the manuscript, with minimal revision of its adequacy to editorial standards and typing.

Reviewer 2 Report

This manuscript is hard to be considered a completed study. It looks like a doctoral proposal or a funding proposal. The authors claimed that this is the protocol for extracting nursing skill mix, and I agree with that. But the study does not provide any results or evidence that the suggested protocol is better than the methods in the previous studies. The authors provide the limitations of the previous studies in the introduction and the detailed method section. But those contents are good to be the introduction and the method section for a study but not good enough to be a completed study. Most of the contents seem to be the detailed information of future data collecting process for a doctoral proposal, and readers cannot know findings and implications of this study because we don't have any results from the data collecting process. If the study is completed with the suggested process, the study will be very worthy to be published. But in this current format, all most everything in the manuscript is assumptive. Plus, the authors do not provide any implications of the study in the discussion, and the conclusion section is not conclusive, persuasive, and informative. It’s convinced that this study will be very empirical and informative and “will provide evidence of the feasibility of extracting routinely collected administrative patient and nurse data to calculate measure of nursing skill mix”(p. 8 in the manuscript) after the authors conduct the study with the approaches suggested in the manuscript.

Reviewer 3 Report

It is a very interesting research proposal that could provide a good guide for future studies in the field of mental health.

Some minor comments:

It would be good to review and modify the wording of both the title and the paragraphs where the objective of the study is mentioned, for example, lines 17 to 19 and 344 to 349, so that they are more consistent with each other.

As they are factors that could affect the results of future studies that use this protocol to explain patient readmissions based on the skills of the nurses and to be able to control them as intervening variables, it would be interesting to add to what is already included: the type of comorbidity psychiatric disease, the type of physical comorbidity, the consumption of toxic substances (type, onset, quantity, etc.), duration or date of the first diagnosis of the psychiatric disease, pharmacological treatment, and of the nurses, the length of work experience (months, years) and the type of contract (full-time or part-time).

In addition, because the study period (data collection) is included within the period of the COVID19 pandemic, variables such as isolation/confinement, vaccination (dose), infections (no, yes – number), should also be taken into account as an intervening factor for both patients and nurses. 

If these data could not be extracted from the databases or are difficult to obtain directly, it would be good to mention it within the limitations.

Round 2

Reviewer 2 Report

The manuscript has been improved a lot with the revision, and the current format of the manuscript can be accepted.